# OpenReview forum: "Can't hide behind the frame:  Disentangling goal & framing for detecting LLM jailbreaks"
_ICLR.cc/2026/Conference — ICLR 2026 Conference Withdrawn Submission_

### Official Review · Reviewer_ECj4 · 2025-10-26

**Soundness:** 3
**Presentation:** 3
**Contribution:** 3
**Rating:** 6
**Confidence:** 4

**Summary:**

This paper analyzes prompt-based automatic iterative refinement jailbreak attacks from the perspective of *framing* and *goal*. It constructs a dataset by combining different framings and goals, and exploits a latent layer of the target LLM to disentangle the features corresponding to the framing and the goal of input prompts. Based on the differences between the framings of jailbreak prompts and benign prompts, the paper proposes an anomaly detection-based jailbreak defense method. Experiments on different datasets and models show improvements compared to existing defenses.

**Strengths:**

1. The perspective of this paper is interesting, novel, and potentially correct. It divides jailbreak prompts into *framing* and *goal* components, extracts the true goal through feature disentanglement, and uses this for detection. This idea has been successfully applied in malicious traffic detection and backdoor detection.
2. The experiments are thorough, and the ablation studies are well designed, effectively supporting the validity of each component of the proposed method.

**Weaknesses:**

1. There are issues with the theoretical proof section. Overall, Sections 4.1–4.5 demonstrate that the model can characterize a specific dataset and that when the sampling size reaches the lower bound, the model can learn all disentangled information about the *goal* and *framing* contained in that dataset. However, this is not equivalent to theoretically guaranteed robustness. For instance, in adversarial robustness methods based on random smoothing, the guarantee applies to the robustness of any input in the entire input space. In contrast, the theoretical analysis in this paper only guarantees robustness for the training dataset under specific conditions, while the robustness for test data depends on the model’s generalization ability and lacks theoretical support. Therefore, I believe this theoretical section is not the main contribution of the paper, yet it occupies two pages, which makes the paper more difficult to read.
2. The paper only reports the detection success rate on the test datasets but does not provide the attack success rates of those datasets themselves. For some datasets, the attack success rate on Llama2-7B is already below 5%; in such cases, the significance of detecting jailbreak attacks is reduced (although it still has some value). Moreover, for certain models, the F1 score of the proposed method drops significantly (e.g., in Table 2), and this issue deserves further analysis.

**Questions:**

Overall, I think this is an interesting paper that proposes a new analytical perspective on a popular topic and could inspire future research in this field. However, the writing can still be improved, especially in the theoretical section. I suggest moving the theoretical proofs to the appendix to improve the readability of the main text.

---

### Official Review · Reviewer_NZQd · 2025-10-30

**Soundness:** 4
**Presentation:** 4
**Contribution:** 3
**Rating:** 6
**Confidence:** 4

**Summary:**

This paper proposes a framework for disentangling semantic factors, specifically goal and framing, in the internal representations of LLMs to detect adversarial jailbreak attacks. The approach is instantiated through three main contributions: (i) a new PAIR+Framing dataset comprising controlled variations of goal/framing in adversarial prompts, (ii) the ReDAct module for disentangled representation learning from LLM activations, and (iii) the FrameShield detector that leverages framing representations for jailbreak detection.

**Strengths:**

**1. Originality:** The goal-framing decomposition provides a lens for understanding PAIR attacks, grounded in framing theory. This conceptual framework is a genuine contribution.

**2. Quality:** The empirical work is strong—SOTA performance across 8 LLMs with comprehensive experiments and proper methodology.

**3. Clarity:** Generally well-structured with good figures and comprehensive appendices.

**Weaknesses:**

**1. Derivation inconsistency:** The paper claims Lemma 4.2 provides theoretical justification for the sample complexity needed to satisfy Proposition 4.1's requirements. However, Lemma 4.2 **only** guarantees: Each factor value a ∈ A and b ∈ B appears at least once in the sample with probability 1-2δ when n ≥ (1/p_min) log(|A|/δ ∨ |B|/δ) and Proposition 4.1 requires: Each a ∈ A appears in at least one pair in P_A, and each b ∈ B appears in at least one pair in P_B. This gap will make the claimed sample complexity severely underestimated.

**2. Lack of evaluation on real jailbreaking attacks:** While demonstrate good performance on author's dataset, it is essential to evaluate the methods on real PAIR style attacks, such as PAIR (Attack) [1], DrAttack [2], Puzzler [3], Persuasion [4] and etc.

[1] Jailbreaking Black Box Large Language Models in Twenty Queries

[2] DrAttack: Prompt Decomposition and Reconstruction Makes Powerful LLM Jailbreakers

[3] Play guessing game with llm: Indirect jailbreak attack with implicit clues

[4] How Johnny Can Persuade LLMs to Jailbreak Them: Rethinking Persuasion to Challenge AI Safety by Humanizing LLMs

**Questions:**

1. How are the multiple objectives balanced and tuned through the weighting scheme? Could the authors provide details on the weight selection process and discuss any observed trade-offs between objectives?

2. Could the authors provide ablation studies or qualitative analysis on failure modes of FrameShield? Specifically, are there particular goal types or framing domains where the method is less effective?

3. Has FrameShield been evaluated on over-refusal benchmarks such as XSTest [1] and OR-Bench [2] to assess whether it introduces false positives on benign queries? This would help characterize the method's precision and identify any tendency toward over-sensitivity.

[1] XSTest: A Test Suite for Identifying Exaggerated Safety Behaviours in Large Language Models

[2] OR-Bench: An Over-Refusal Benchmark for Large Language Models

---

### Official Review · Reviewer_tejk · 2025-10-30

**Soundness:** 3
**Presentation:** 3
**Contribution:** 2
**Rating:** 2
**Confidence:** 4

**Summary:**

This paper takes an approach to (1) jailbreak detection and (2) disentengling how LLMs represent different features in their prompts such as framing vs. request. In a series of experiments, the paper (1) introduces a dataset construction method that augments data to disentengle jailbreak requests and frames, (2) fits contrastive models to distinguish different frames and requests, and (3) uses these models as a jailbreak detection method.

**Strengths:**

S1: I think that contrastive pair construction and disentanglement are genuinely clever and good. I can see how this method has roots in past work on mechinterp, but as far as I know and in my opinion, this seems genuinely interesting and distinct.

S2: I'm glad that ReDAct doesn't use a linear probe as many mechinterp papers would attempt to do.

S3: I think that the paper could benefit from more examples and explanation in section 4 but that it's well and clearly written overall.

**Weaknesses:**

W1: As I read the abstract, I am immediately unsure of why PAIR is highlighted in particular. PAIR is good but not really SOTA. Manual jailbreaks have always been better. Meanwhile, multiturn methods like CASCADE seem empirically stronger. I am expecting to be inherently suspicious of any evaluations for attack detection that only focus on one attack. Meanwhile, maybe I am just too sleep deprived, but I am struggling to parse the abstract given undefined terms and sentence length/structure in it.

W2:  The generality of the framework seems to be limited since (1) goals can't always be cleanly conceptually separated from framing, and (2) not all jailbreaks are semantically interpretable.

W3: I think that the paeper is overclaiming, and I would recommend rewriting it to (1) use more than just PAIR and do more general types of experiments with an expanded set of semantically interpretable jailbreaks and (2) drop FrameShield as a central contribution -- it's a demo but not shown to be valuable in practice because the paper doesn't evaluate it on jailbreaks other than PAIR.

W4: I am usually pretty suspicious of probing or mechinterp experiments that don't show generalization to things that the method was not fit using. Tests on non-pair prompts or jailbreaks seem necessary.

W5: Unless I'm missing something, semantic factor is not defined, and section 4 is unintelligible starting at the first paragraph. I don't know how the first sentence of 4.1 can make sense if the semantic factors are not known. I don't think the sentence on line 174-176 makes sense either. When section 4 gets revisited, some examples should at least be used.

W6: Please correct me if/when I'm wrong here. But my initial reaction to section 4 is mostly a shrug. My thinking is that "So what about any of the propositions. We don't have any guarantees that the decoder's job is tractable from a statistical learning perspective unless we just assume that it is. The representations might not be friendly." I am inclined to wonder if section 4 overformalizes. If we assume that two independently modulatable semantic factors are represented in model latents, then of course we ought to be able to contrastively probe for them using the right data. There is probably something I'm not understanding, but I am not seeing any value in section 4 beyond those assumptions (which I think are baked into section 4 anyway).

W7: I think that section 7 conflates how easy it is to fit ReDAct with "where the representations corresponding to X are truly stored." I think this is a little sus. Mechinterp research has a history of claims saying that "the feature is stored here" only for subsequent experiments ablating "here" to find that the model still processes the feature.

**Questions:**

See weaknesses

---

### Official Review · Reviewer_Kpjt · 2025-11-01

**Soundness:** 2
**Presentation:** 3
**Contribution:** 2
**Rating:** 2
**Confidence:** 3

**Summary:**

This paper proposes a framework for detecting PAIR-type jailbreak attacks by disentangling "goal" and "framing" semantic factors in LLM representations. The authors introduce PAIR+Framing (an augmented dataset), ReDAct (a disentanglement module), and FrameShield (an anomaly detector), achieving up to 21% accuracy improvements over baselines.

**Strengths:**

Strengths

1. Novel theoretical framework: To my knowledge, the connection between framing theory from behavioral economics and jailbreak detection is creative and well-motivated. The formal treatment in Section 3/Appendix B provides useful conceptual grounding. Authors tried to formalize the theory around PAIR attacks.

2. Empirical results: Authors show that they can achieve good performance across multiple LLM families (Llama, Vicuna, Mistral, Qwen) with consistent improvements over JBShield.

3. Theoretical contributions: Propositions 4.1-4.5 provide formal guarantees for the disentanglement framework, though with some caveats (see weaknesses).

4. Model independence: The approach works across different LLM architectures without fine-tuning the base models.

5. Interpretability insights: The layer-wise analysis (Section 6), revealing differential concentration of goal/framing signals, is interesting from the mechanistic interpretability side.

**Weaknesses:**

**Major Weaknesses**

1. Severely Limited Scope

My main concern with this paper is the limited scope. The framework only handles binary semantic factor pairs. The authors acknowledge this (Section 7), but it's a fundamental limitation. Real prompts likely involve multiple interacting semantic dimensions (tone, context, persona, etc.). The claim of "broader applications" is undermined by this restriction. Furthermore, there are “past tense” jailbreaking attacks (https://openreview.net/forum?id=aJUuere4fM), which are somehow related to PAIR-style attacks could have been investigated.

2. "Controlled Leakage" is Poorly Formalized

Remark 4.4 introduces "controlled leakage" as essential but never rigorously defines it. The bound in Proposition 4.5 (Δ ≤ ε/λ_orth) is:

* Too loose to be practically useful
* Provides no guidance on setting λ_orth
* Doesn't characterize what "helpful leakage" means formally

This is a critical gap since complete disentanglement is explicitly not the goal.

3. Vulnerability to Adaptive Attacks

No adversarial evaluation is provided. Critical questions remain unanswered:

* What if attackers know FrameShield exists and optimize framings to evade detection?
* Can adversaries craft "benign-looking" framings for harmful goals?
* The anomaly detection approach (95th percentile threshold) seems easily gameable

I think this is a significant omission defense method.

4. Generalization Claims Not well Supported

My second main concern (also related to the scope) is regarding the evaluation.

* Only evaluated on PAIR-type attacks (iterative paraphrasing)
* No evaluation on: GCG, AutoDAN, other semantic jailbreaks to see the transferability
* Table 3's "OOD" evaluation only varies goal categories, not framing styles.
* Will this detect completely novel framing strategies?

5. Dataset Generation Quality Unclear

PAIR+Framing relies heavily on GPT-4.1 generation:

* What quality control was performed?
* How many generations were rejected/filtered?
* Could generation artifacts create spurious patterns?
* Inter-rater reliability not reported

The 10 variations per prompt × 2 dimensions could amplify systematic biases.


**Minor issues**

* Proposition 4.1:  I think the sufficiency result is straightforward (essentially graph connectivity), and doesn't provide insight into sample complexity.

* Proposition 4.3: Asymptotic sufficiency requires τ→0 and infinite data. The finite-sample guarantees would be more useful.

* Incomplete Disentanglement: able 1 shows η² values of only 0.22-0.41 on diagonal:This means representations explain <41% of variance in their intended factors; Off-diagonal values (0.12-0.19) are concerningly high. Is this level of disentanglement sufficient? No ablation studies varying λ_orth.

* Computational Cost Downplayed: Training: ~4 hours/layer × 32 layers = 128 GPU-hours on H100s; therefore "Minimal computational overhead" is misleading.

* Statistical Issues:  The critical layer selection (FrameShield-Crit) uses Cohen's d, assuming normality. This assumption is not validated.

**Questions:**

1. How does FrameShield perform against adaptive attacks that optimize framing to evade detection?
2. Can you provide inter-annotator agreement on goal vs. framing boundaries?
3. What is the false positive rate on diverse benign prompts (creative writing, technical documentation, etc.)?
4. How does performance degrade with fewer training pairs?
5. Can the framework extend to 3+ factors? What breaks theoretically?
6. Why not compare to fine-tuning based defenses or circuit breakers (Zou et al., 2024)?

---

### Author Response · Authors · 2025-12-03
**Main Takeaways**

The reviewers’ recognition of the merits of the work is encouraging, and their feedback on the shortcomings is constructive. We are, however, withdrawing, since we need to revise the presentation and supplement our base dataset, and hence rerun the full experimental pipeline, making the required revision beyond this rebuttal phase. We would like to summarize the main takeaways below.

The positive reception of this work is very encouraging. In particular, the reviewers have highlighted the strengths along multiple dimensions:

- **Conceptual novelty and potential impact:**
All reviewers emphasize the novelty of decomposing jailbreak prompts into goal and framing and grounding this in framing theory. [Kpjt] calls the link to framing theory *“creative and well-motivated,”* [NZQd] describes the decomposition as a *“genuine contribution,”* [ECj4] notes that the perspective is *“interesting, novel, and potentially correct,”* and [tejk] finds the contrastive pair construction and disentanglement *“genuinely clever and good”* and *“genuinely interesting and distinct.”* Reviewers also recognize the potential impact, e.g., [ECj4] concludes that it *“could inspire future research.”*

-  **Representation-level approach & mechanistic interpretability:**
Reviewers appreciate that the paper proposes a representation-level solution, and links jailbreak detection to mechanistic interpretability, using innovative techniques that rely on well-established tools. [tejk] notes that they are *“glad that ReDAct doesn’t use a linear probe,”* while [Kpjt] highlights the layer-wise analysis as *“interesting from the mechanistic interpretability side.”*

-  **Theoretical grounding:**
The reviewers recognize the value in the formalization of our framework, though they also point out the need for improvements, which we agree with and will resolve in our revision. [Kpjt] points to *“theoretical contributions”* and notes that the propositions provide *“formal guarantees for the disentanglement framework”*.

- **Empirical strength:**
Reviewers stress that the empirical study is substantial and carefully executed, even as they point out the issue with being PAIR-centric, which we agree to improve. [Kpjt] writes that we *“achieve good performance across multiple LLM families,”* [NZQd] characterizes the empirical work as *“strong—SOTA performance across 8 LLMs with comprehensive experiments and proper methodology,”* and [ECj4] notes that *“the experiments are thorough, and the ablation studies are well designed.”*

We believe these points reflect the merits of our paper. That being said, we acknowledge the shortcomings the reviewers have pointed out, as we discuss next.

Considering the reviews, the main shortcoming of the paper is that the presentation and empirical design make the work appear centered around PAIR. Our framework is model- and attack-agnostic as long as the attack relies on manipulating the framing of a benign goal (as formally described in Section 3). The PAIR dataset is meant as an illustrative testbed for instantiating this framework, not to determine the limits of its applicability. We appreciate this feedback and agree that the current manuscript’s over-emphasis on PAIR misrepresents its role. While we stated that the framework is not limited to PAIR, we acknowledge that several factors contributed to this shortcoming, including the disproportionate focus of the writing on PAIR, the naming of our dataset, and the lack of diversity in base datasets. We take responsibility for this and will make it a top priority in future submissions to reduce PAIR-centric presentation and empirical design to better reflect the broader scope.

Other shortcomings raised by the reviewers include missing details in our formalization and the lack of analysis of our PAIR+Framing dataset. We appreciate these questions. In particular, in revising the paper we will:

- clearly define all concepts and tools formalized and used in Section 4;

- revise the presentation to clarify that our specific jailbreak detection method is not intended as a plug-and-play centerpiece, but as a demonstration of how the framework can lead to improvements in LLM safety;

- include a descriptive analysis of our contrastive pairs dataset.
Together with revising the draft and experiments to address the PAIR-centric narrative, as discussed above, we believe these changes will resolve the identified shortcomings.

To conclude, we see this review round as a clear indication that the core contribution is promising, and as an opportunity to resolve the remaining issues. We are grateful for the reviewers’ detailed and constructive feedback, and we are committed to using it to strengthen the paper for a resubmission.

---

### Note · Authors · 2026-01-08

**Comment:**

We thank the reviewers and AC for their time and thoughtful feedback.

We are encouraged by the positive reception of our core ideas and contributions. The reviewers **unanimously acknowledged the novelty and significance of the problem we target and our approach towards a systematic solution**, while pointing out some shortcomings. In our comment below, we briefly discussed the main points raised.

Although the reviews confirm our framework's novel contributions, the main shortcoming requires a substantial revision of the paper’s writing. As a result, we have decided to withdraw at this point. That said, we find the reviews highly positive and promising, and are committed to strengthening the paper using this constructive feedback for resubmission.

Once again, we thank the reviewers and the AC for their valuable time. We are confident that this feedback will help us prepare a manuscript that resolves the identified shortcoming and better presents the merits of the paper, as already highlighted by the reviewers.

**Withdrawal Confirmation:**

I have read and agree with the venue's withdrawal policy on behalf of myself and my co-authors.